# Diversity and function of soluble heterodisulfide reductases in methane-metabolizing archaea

Xingyu Lyu,[1] Hang Yu,[1] Yahai Lu[1]

**ABSTRACT** Soluble heterodisulfide reductase subunit A (HdrA) is an ancient protein central to energy metabolism, facilitating the recycling of intermediates in methane metabolism and performing flavin-based electron bifurcation for energy conservation. In this study, we investigated the functional diversity and evolutionary dynamics of HdrA in methane-metabolizing archaea. An analysis of 1,152 HdrA sequences from 624 genomes revealed that HdrA diversified through internal domain modifications, resulting in 28 distinct classes and 4 major types (types I, Ia, II, and III). Functional genes in HdrA gene clusters revealed variations in mid-potential electron donors, including NADH, $F_{420}H_2$, $H_2$, and formate. Two major types of HdrA have not previously been studied in detail. Type II HdrA resulted from a fusion of two different classes of type I HdrA. Particularly, a consistent gene cluster containing type II HdrA, molybdopterin oxidoreductase, and $F_{420}$ dehydrogenase was identified in anaerobic methane-oxidizing archaea and methanogens. Protein sequence and structural predictions suggested that the molybdopterin oxidoreductase protein had lost its catalytic function, and $F_{420}H_2$ served as the mid-potential electron donor or acceptor for the Hdr protein complex. This gene cluster may expand to include additional type I HdrA and HdrD, potentially supporting two electron bifurcation events to lower electron potential for ferredoxin reduction. Type III HdrA, with an inserted GltD domain compared to type I HdrA, appears to have altered the electron transfer route and may use NADH as its mid-potential electron donor or acceptor. The remarkable functional flexibility of HdrA likely helps methane-metabolizing archaea adapt to diverse anaerobic environments.

**IMPORTANCE** All methanogenic archaea use heterodisulfide of coenzymes M and B as the terminal electron acceptor. In anaerobic methane- and alkane-oxidizing archaea, the reverse reaction occurs. The cycling of heterodisulfide is vital to the energy conservation of these anaerobic microorganisms. Soluble heterodisulfide reductase is an ancient protein fulfilling this function via flavin-based electron bifurcation or confurcation. Despite being present in the vast majority of methane- and alkane-metabolizing archaea, the diversity and evolution of this key protein have not been investigated. This study reveals substantial domain variation and structural changes in the key bifurcating subunit HdrA in methane- and alkane-metabolizing archaea. The resulting flexibility of HdrA enables the protein complex to vary its interacting subunits and electron carriers based on the organisms' primary metabolism. Our findings shed light on how methane- and alkane-metabolizing archaea thrive in various anaerobic environments, contributing to our broader understanding of carbon cycling and energy conservation.

**KEYWORDS** heterodisulfide reductase, HdrA, electron bifurcation, electron confurcation, energy conservation, FBEB, methane metabolism, methanogen, anaerobic methane-oxidizing archaea, ANME, dimethyl sulfoxide reductase, molybdopterin oxidoreductase

**Peer Reviewers** Nana Shao, University of Georgia, Athens, Georgia, USA; Kylie Allen, Virginia Polytechnic Institute and State University, Blacksburg, Virginia, USA

Address correspondence to Hang Yu, yuhanghank@pku.edu.cn, or Yahai Lu, luyh@pku.edu.cn.

The authors declare no conflict of interest.

See the funding table on p. 13.

Methane is a potent greenhouse gas with a global warming potential 28 times greater than that of carbon dioxide (1). As the final product of anaerobic organic matter decomposition, microorganisms including methanogenic archaea and anaerobic methane-oxidizing (ANME) archaea play a crucial role in its cycling in anoxic environments (2, 3). The enzyme methyl-coenzyme M reductase (Mcr) is central to the methane cycle, facilitating methane production in methanogens by reducing methyl-coenzyme M ($CH_3$-S-CoM) with coenzyme B (HS-CoB) to produce heterodisulfide (CoB-S-S-CoM) and methane (4). To sustain this methanogenic process, CoB-S-S-CoM must be reduced back to HS-CoB and HS-CoM by the heterodisulfide reductase (Hdr) complex (3, 5). In ANME archaea, the reverse process occurs during anaerobic methane oxidation, underscoring the interconnectedness of Mcr and Hdr systems for the biological production and consumption of methane.

In addition to methane metabolism, Mcr-like alkyl-coenzyme M reductases (Acr) have been identified across diverse anaerobic environments, where they metabolize short-chain alkanes similarly to how Mcr metabolizes methane (6–8). The Acr-based processes also require the cooperation of Hdr complexes, extending the essential role of Hdr in anaerobic biogeochemical cycles beyond methane (7).

Flavin-based electron bifurcation (FBEB) is the most recently discovered energy-conserving mechanism in anaerobes (9–12). HdrA has the unique ability to facilitate FBEB by coupling the exergonic reduction of high-potential electron acceptor CoB-S-S-CoM and the endergonic reduction of low-potential electron acceptor ferredoxin (Fd) to the oxidation of mid-potential electron donor such as hydrogen (13, 14) (Fig. 1A). The reduced Fd produced by this mechanism is essential for various cellular reactions, including activating $CO_2$ in hydrogenotrophic methanogens (15); (Fig. 1A). The reverse process, flavin-based electron confurcation (FBEC), also occurs to produce mid-potential electron acceptor, as shown by comparative genomic and biochemical studies (16, 17). This mechanism enables the coupling of endergonic and exergonic reactions in reverse methanogenesis. As these organisms evolved, cytochrome-containing hydrogenotrophic, acetoclastic, and methylotrophic methanogens developed a membrane protein complex HdrDE for CoB-S-S-CoM reduction, reducing their dependence on the soluble HdrABC complex (18–22). Yet, the majority of these methanogens still retain the soluble HdrABC complex (18, 23). In the complex, HdrB and HdrC form a functional unit responsible for catalysis and electron transfer. HdrB contains an unusual [4Fe-4S] cluster, serving as the key catalytic site for CoM-S-S-CoB reduction (24). HdrC facilitates electron transfer through its two conventional [4Fe-4S] clusters (25). The complete electron transfer pathway begins with HdrA receiving electrons from electron donors, transferring them via flavin adenine dinucleotide (FAD) and [4Fe-4S] clusters to HdrC, which then relays these electrons to HdrB's specialized [4Fe-4S] cluster for CoM-S-S-CoB reduction (3, 26).

HdrA proteins contain two key cofactors: iron-sulfur cluster for electron transfer and FAD similar to thioredoxin reductase for FBEB (9, 12, 27). Previous studies have noticed that HdrA has diversified by modifying its cofactors in methanogenic (17) and ANME archaea (16). Yet, there has been no comprehensive study examining the functional diversity and evolutionary patterns of the HdrA gene across microorganisms containing Mcr and Acr. To address this knowledge gap, we analyzed 1,152 soluble Hdr subunits in 624 methane- and alkane-metabolizing archaea genomes. By delving into the diversity, evolution, and structure of HdrA, our results uncovered new types of this protein and shed light on how methane- and alkane-metabolizing archaea connect their primary metabolism, via HdrA, to the key cellular electron carriers.

## RESULTS

### Internal domain diversity and classification of HdrA

To explore the diversity of soluble Hdr, we analyzed 624 archaeal genomes containing *mcrA*- and *mcrA*-like genes (Table S1). The majority of these methane- and alkane-metabolizing archaea (559/624) contained *hdrA*. Two methanogen

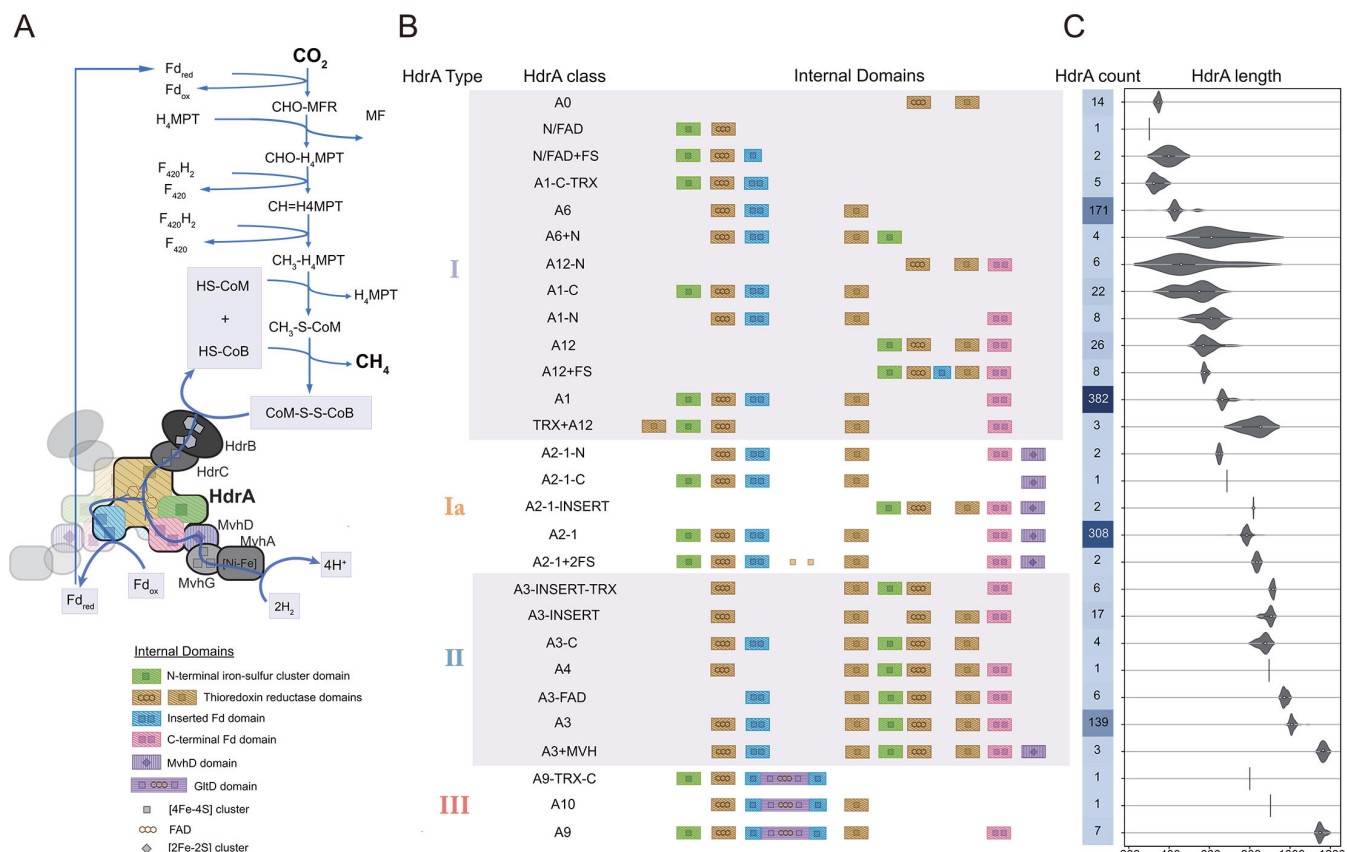

**FIG 1** Classification of HdrA based on protein primary structure. (A) $CO_2$ reduction and energy conservation mechanism in methanogenic archaea. The figure illustrates the enzymatic pathway converting $CO_2$ to $CH_4$, with the final step involving the reduction of the heterodisulfide (CoM-S-S-CoB) into their thiol forms (HS-CoM + HS CoB). The HdrABC-MvhAGD complex plays a central role in this process, employing an FBEB mechanism mediated by HdrA. This mechanism facilitates the simultaneous distribution of electrons from $H_2$ oxidation, catalyzed by MvhAGD ([NiFe]-hydrogenase), to two electron acceptors with differing potentials: the high-potential CoM-S-S-CoB (via HdrB) and the low-potential Fd (via HdrA). This bifurcation mechanism not only provides electrons for the reduction of $CO_2$ but also generates $Fd_{red}$, which drives the initial reduction of $CO_2$ to formyl-methanofuran. This process is fundamental to the energy metabolism of methanogenic archaea. The cartoon representation of HdrA highlights its key functional domains and associated cofactors, as depicted in the legend. Blue arrows indicate the flow of electrons and the direction of the enzymatic reactions in the pathway. (B) Schematic representation of 28 classes in 4 major HdrA types (types I, Ia, II, and III). Identified internal domains across different HdrA classes are aligned based on sequence similarity. Compared to type I, besides variations in iron-sulfur clusters, type Ia has a fused MvhD domain at the C-terminus potentially transferring electrons or reacting with $F_{420}H_2$, type II includes an additional thioredoxin reductase domain potentially bifurcating electrons, and type III has an inserted GltD domain potentially reacting with NAD(P)H. (C) The count and length distribution of HdrA proteins across 624 genomes of methane- and alkane-metabolizing archaea. The heatmap illustrates the gene counts of *hdrA* across various classes. The violin plot represents the average protein length for each class.

groups, Methanonatronarchaeia and Verstraetearchaeota/Methanomethyliales, have been reported to lack HdrA and may utilize membrane-bound HdrD or HdrBC to reduce CoM-S-S-CoB (28, 29). Hydrogen-evolving membrane-bound hydrogenase complex or the energy-converting hydrogenase B (Ehb) complex could generate reduced ferredoxin required for anabolic reactions (8). In our analysis of other methanogens like Methanocaldococcus that lack HdrA, we identified partial *hdrA* sequences in their genomes. This absence of complete *hdrA* appears to result from the presence of selenocysteine residues, which are misinterpreted as stop codons in gene prediction tools. The resulting 1,152 HdrA protein sequences (Table S2) showed substantial sequence length variation from 302 to 1,182 amino acids due to differences in their recognized functional domains (Fig. 1).

The most common HdrA proteins, including the two that have been structurally characterized (24, 30), contain five conserved domains: an N-terminal iron-sulfur cluster

domain, thioredoxin reductase domains binding an FAD and an iron-sulfur cluster, an inserted Fd domain, and a C-terminal Fd domain, both binding two iron-sulfur clusters. Our analysis revealed that the number of conserved domains in HdrA showed substantial variation and could differ from 2 (A0) to 8 (A3+MVH; Fig. 1B). We classified the HdrA into 28 distinct classes based on the domain variations. These classes were further grouped into four major types: types I, Ia, II, and III, based on their thioredoxin reductase domains. The HdrA classification system builds upon established nomenclature (16) and is structured around three primary types based on structural characteristics. Type I centers on A1, with A0 representing the minimal Hdr-like proteins in sulfur oxidizers, and variants named by domain modifications (e.g., A1-C lacks the C-terminal Fd domain). Type Ia derives from A2-1 (formerly HdrA2) (17, 31, 32), with subclasses defined by structural variations. Type II is founded on A3 and A4, while type III is based on A9 and A10 (16), with their respective subclasses categorized by domain composition relative to these foundational structures.

Type I, having one thioredoxin reductase domain and varying numbers of iron-sulfur clusters, was the most prevalent type that was previously biochemically and structurally studied (13, 20, 24, 30, 33). It included 13 classes. The most common classes were HdrA1 (382 sequences) with five domains and HdrA6 (171 sequences) which has two fewer domains (N-terminal iron-sulfur cluster and C-terminal Fd domains) compared to HdrA1 (Fig. 1B and C). Type Ia was a subtype of type I, distinguished by the fusion of MvhD (an electron transfer protein with one [2Fe-2S] cluster) at its C-terminus. It included five classes. The most common class was HdrA2-1 (308 sequences) with domains similar to HdrA1 of type I. Type II HdrA contained an additional thioredoxin reductase domain and also exhibited variations in the number of iron-sulfur clusters. It included seven classes (176 sequences). The most common class was A3 (139 sequences). Type III HdrA, in contrast to types I and II, contained an inserted GltD domain. GltD is the small subunit of NAD(P)$^+$-dependent glutamate synthases, and this subunit catalyzes reactions with NAD(P)H (34). The inserted GltD domains in type III HdrA contained a GXGXXG motif rather than GXGXXA, which was characteristic of the Rossmann fold and indicated preferential binding of NAD$^+$ rather than NADP$^+$ (35–38). Type III HdrA was the least represented with only nine sequences. Besides the insertion of the GltD domain, its domain pattern was most similar to HdrA1 of type I. Comparative domain analysis between the HdrA types revealed that certain classes, such as A12 from type I, shared domain similarities with regions near the C-terminus of type II. These domain patterns led us to wonder how they evolved.

## Phylogeny and structure of HdrA proteins

Given the substantial variations in HdrA sequence length and domains, we focused on the sequence region most relevant to its FBEB/FBEC function for evolutionary and structural analysis. To define this sequence region, we initially reduced our data set to 401 representatives by clustering based on sequence similarity (80% amino acid similarity cut-off). After clustering, we removed regions with low information content or high variability based on the multiple sequence alignment (Fig. S1). Only the highly conserved sequence regions with strong evolutionary signals containing 252 amino acids were used in our phylogenetic analysis (see Materials and Methods for selection details).

The phylogeny of HdrA showed a mosaic pattern for its 28 classes, but multiple evolutionary events could give rise to its four major types (Fig. 2; Fig. S2). Type I, likely the most ancient type, was separated into multiple phylogenetic clades, each containing different HdrA classes (Fig. 2). The simplest class A0 of type I occurred in multiple clades, possibly indicating domain addition and deletion to HdrA. Type Ia was likely the result of type I fusing with MvhD. Despite our phylogenetic analysis excluding the fused MvhD region of type Ia HdrA, types I and Ia still formed distinct phylogenetic branches, indicating that multiple fusion events of type I produced type Ia. Because type II HdrA contained two thioredoxin reductase domains that could have different evolutionary

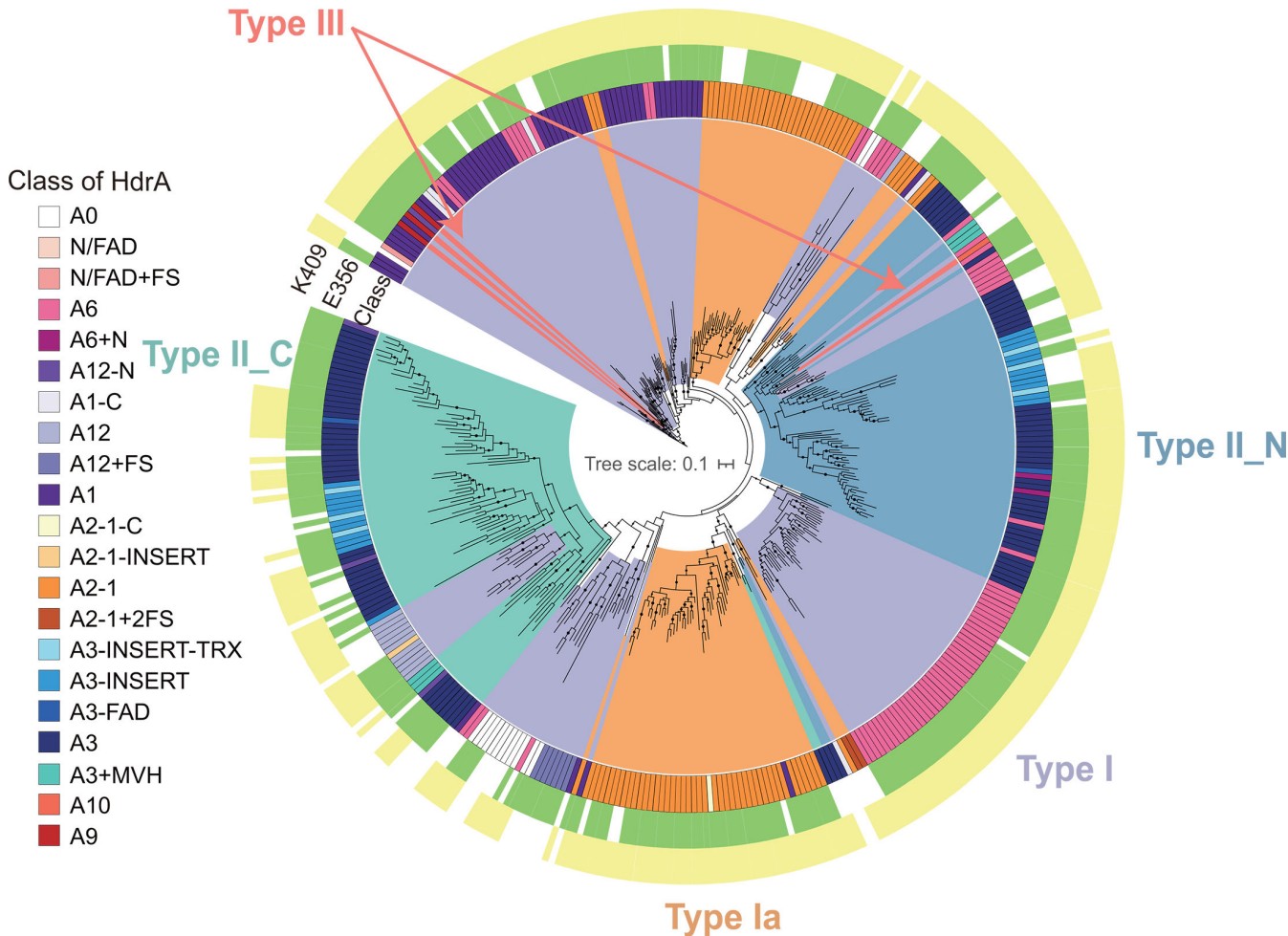

**FIG 2** The presence of key amino acid residues in the core region of different HdrA classes and types. The phylogenetic tree of the conserved core regions (252 amino acids) shared between all HdrA types: type I (purple), type Ia (orange), type II (blue and green), and type III (red). Type II sequences with two thioredoxin reductase domains are analyzed in two segments: type II_N (blue) and type II_C (green). The innermost ring represents the (21/28) classes of HdrA (Fig. 1). The presence of key residues K409 and E356 for stabilizing FAD, along with other residues involved in FAD binding and stabilization. K409 and E356 refer to the corresponding residues in HdrA from *Methanothermococcus thermolithotrophicus* (WP_018154264).

histories, we split the type II HdrA sequences and analyzed the core sequence region of their N-terminus (type II_N) and C-terminus (type II_C) separately. Indeed, type II_N and type II_C regions formed independent phylogenetic branches. Type II_N likely originated from a fusion of classes A6 and A12 of type I HdrA due to the presence of and domain similarity to these type I classes within type II_N and type II_C clades. The nine sequences and three classes of type III HdrA appeared toward the tip of the phylogenetic tree. After removing the inserted GltD domain of type III HdrA, the remaining domain pattern matched the most related HdrA class in their respective clade (type III A9 and A9-TRX-C evolved from type I A6 and type III A10 evolved from A1), indicating that two separate insertion events gave rise to the type III HdrA in methane-metabolizing archaea (Fig. 2). This was supported by the observation that type III A9 and A9-TRX-C were exclusively in ANME-1, while type III A10 was found in Bathyarchaeota.

Previous structural studies of HdrA suggested that two amino acid residues, lysine (K) and glutamic acid (E), contributed to stabilizing negatively charged intermediates by interacting with the N5 position of FAD (24, 39, 40), e.g., K409 and E356 in HdrA from *Methanothermococcus thermolithotrophicus* (24). The presence or absence of these conserved residues was proposed as a critical feature distinguishing the electron-bifurcating from the non-bifurcating proteins (24, 40). We analyzed these two key residues

and found both to be conserved across most HdrA proteins (Fig. 2). Specifically, only 61 sequences (15.2%) and 87 sequences (21.7%) out of the total 401 sequences analyzed lacked K409 and E356, respectively. Interestingly, both K409 and E356 were conserved in all type III sequences and 75% of type II sequences, suggesting that both thioredoxin reductase domains in type II as well as the thioredoxin reductase domain in type III HdrA could be capable of performing FBEB/FBEC.

## Potential FBEB/FBEC functions of HdrA complex based on gene cluster analysis

HdrABC and MvhADG are known to form a protein complex for FBEB in methanogens without cytochrome (24). This complex may further form super-complexes with other functional proteins to facilitate the electron transfer reaction. In our previous analysis of *Methanocella conradii*, the clustering of *mvhD*, *hdrABC*, and *fwdABCDFG* genes led us to propose a bifurcating supercomplex consisting of Mvh-Hdr-Fwd, using electrons from hydrogen to simultaneously reduce heterodisulfide to HS-CoM and HS-CoB and $CO_2$ to formylmethanofuran (41). The ability of soluble Hdr to form bifurcating supercomplexes was experimentally validated with the purification of Fdh-Hdr-Fwd and Mvh-Hdr-Fwd (13, 42, 43) as well as structural characterization of Fdh-Hdr-Fmd complexes (30). In this study, we delved into the gene clusters containing *hdrA* to understand the function diversity associated with soluble Hdr in methane-metabolizing archaea. Of 1,152 HdrA sequences, 513 formed gene clusters with other functional genes. Here, we paid specific attention to the functions of types II and III HdrA, which had not been studied in detail previously.

We found that 14% of type II *hdrA* genes were co-located with genes encoding HdrBC, MvhD, $F_{420}$-dependent formate dehydrogenase beta subunit FdhB and a protein in the molybdopterin oxidoreductase family (Molybdop; Fig. 3A). Molybdop is a large family of proteins sharing a similar structural fold, often binding to Mo/W-bis pyranopterin guanosine dinucleotide (Mo/W-bisPGD) cofactor with a range of substrates including formate (FdhA and Fmd/FwdB) (44). In some gene clusters, the gene cluster also included another type I *hdrA* as well as *hdrD* (Fig. 3A). To clarify the potential functions of the Molybdop protein, we performed phylogenetic analysis on all Molybdop protein sequences (also known as MopB/dimethyl sulfoxide reductase/complex iron-sulfur molybdoenzyme/molybdopterin oxidoreductase protein family) in gene clusters with *hdrA* together with representative sequences from a recent MopB sequence collection (Fig. 3B) (44). Molybdop proteins from methane-metabolizing archaea were divided into three distinct groups. Those from *Methanocellales*, *Methanococcales*, *Methanomicrobiales*, and Bog-38 grouped with FwdB/FmdB, while those from *Methanocellales*, *Methanofasti-diosales*, *Methanotrichales*, and *Nezhaarchaeales* grouped with FdhA (Fig. 3B), indicating that they function as formyl-methanofuran dehydrogenase or in formate oxidation, respectively. Interestingly, Molybdop proteins from ANME-2/3, *Methanomethylovorans*, *Methanolobus*, and *Methanosarcina* formed a distinct evolutionary clade that is closely related to, but separate from, FdhG and FhcB (Fig. 3B). The average sequence similarities of FdhG and FhcB to this new Molybdop clade were only 25.24% and 34.20%, respectively, pointing to potential alterations in its function and substrate.

To elucidate the potential role of this new clade of Molybdop proteins, we performed structural prediction and analyzed their active site in detail (Fig. 3C). Previous studies identified a key amino acid cysteine in the active site necessary to coordinate the Mo/W-bisPGD cofactor (45, 46). In formyltransferase/hydrolase complex beta subunit (FhcB), replacing this cysteine with serine rendered the FhcB proteins non-catalytic without Mo/W-bisPGD cofactor (46). In the new clade of Molybdop proteins, the key cysteine residue was absent from all sequences (Fig. S3). Also, we identified two amino acids, proline and phenylalanine, nearby with their functional groups (a pyrrolidine-loop and a phenyl-group) protruding into the active site (Fig. 3C; Fig. S3). Together, these amino acid changes would likely prevent the protein from having a Mo/W-bisPGD cofactor, rendering it non-catalytic. Given that these Molybdop proteins were all in

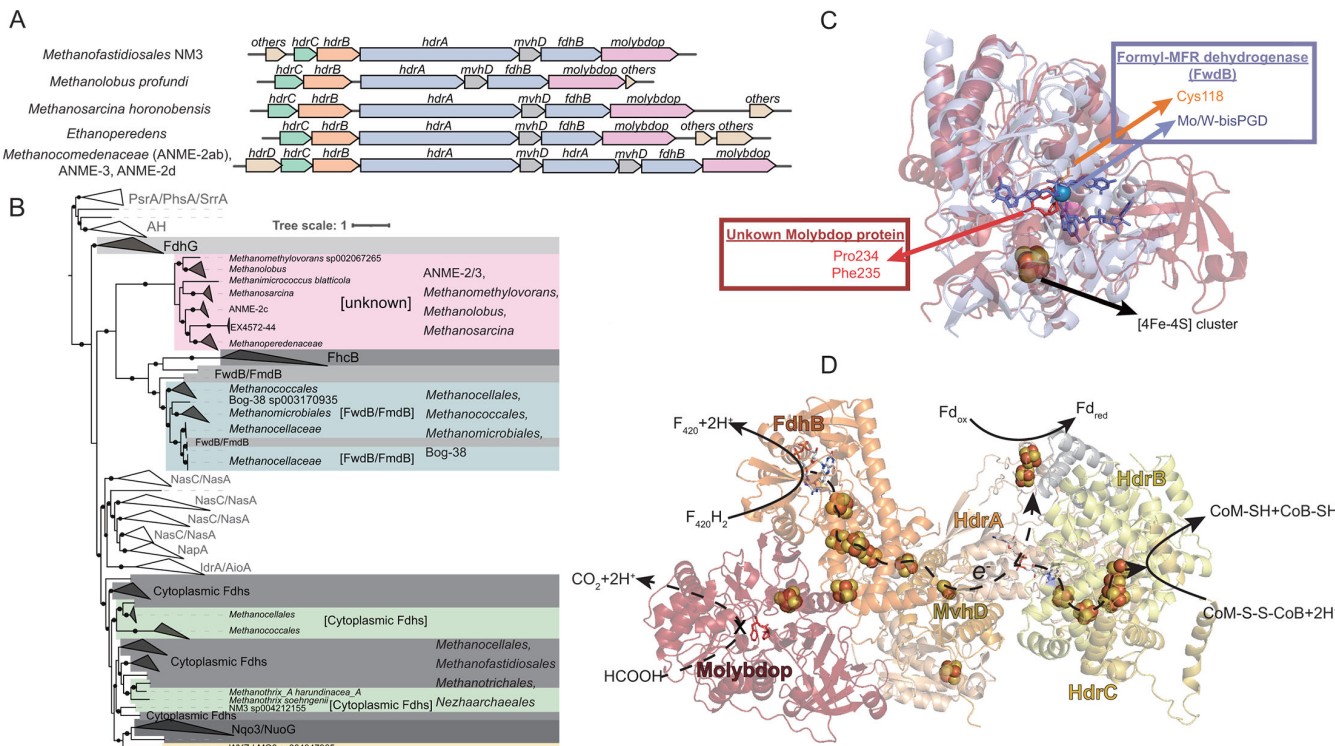

**FIG 3** Analysis of a consistent type II HdrA gene cluster. (A) A consistent type II HdrA gene cluster contains genes encoding an iron-sulfur cluster protein MvhD, an $F_{420}$-dependent formate dehydrogenase beta subunit FdhB, and a protein in the molybdopterin oxidoreductase family (Molybdop). (B) Phylogeny of proteins in the molybdopterin oxidoreductase family, including all Molybdop proteins found in gene clusters with *hdrA*. While some Molybdop proteins with *hdrA* clustered with formylmethanofuran dehydrogenase subunit B Fwd/FmdB (blue) or cytoplasmic formate dehydrogenase alpha subunit FdhA (green), a cluster of Molybdop proteins was identified with unknown function (red). (C) Comparison of a predicted unknown Molybdop protein structure from *Methanolobus profundi* (red) and a characterized FwdB structure from *Methanothermobacter wolfeii* (blue). The Mo/W-bis pyranopterin guanosine dinucleotide (Mo/W-bisPGD) cofactor with its key coordinating Cys118 residue from FwdB is shown. The active site of unknown Molybdop proteins not only lacks the key Cys118 residue but also contains Pro234 and Phe235 residues, preventing Mo/W-bisPGD cofactor from forming. (D) Protein complex structure predicted from the *hdrABC-mvhD-fdhB-Molybdop* gene cluster in *Methanolobus profundi*. The electron transfer pathway potentially for formate (HCOOH) reaction in Molybdop protein is blocked, while the active site and electron transfer pathway for $F_{420}H_2$ reaction in FdhB are viable.

gene clusters with $F_{420}$-dependent formate dehydrogenase beta subunit *fdhB*, we also performed structural prediction and analysis of FdhB alone (Fig. S4) and the whole gene cluster with FdhB, Molybdop, MvhD, and HdrABC together (Fig. 3D). Our analysis found that the *fdhB* and the complex exhibited high similarity to previously characterized HdrABC-FdhAB complex from *Methanospirillum hungatei* (Fig. 3D) (30). Although these complexes contain different protein components, their three-dimensional architectures share remarkable structural similarities, particularly in the core framework and arrangement of functional domains. PyMOL structural alignment shows that 8,664 out of 9,842 atoms could be superimposed with a root mean square deviation of 3.911 Å (values< 4 Å indicates significant structural similarity) (47). The similarity in its structural fold and the presence of key amino acid residues coordinating electron-transfer iron-sulfur clusters were illustrated in Fig. S4. This structural conservation suggests a common evolutionary origin and similar electron transfer mechanisms. Together, the type II HdrA protein complex with FdhB and Molybdop protein is likely capable of using $F_{420}H_2$ as its mid-potential electron donor to perform FBEB reactions for the reduction of CoM-S-S-CoB and Fd, or the reverse for FBEC reactions. Based on our structural analyses, we propose that the Molybdop protein serves as a non-catalytic but structurally supporting component in the protein complex. This proposed role is analogous to FhcB (46), which has been shown to lack catalytic activity but plays an essential structural role in stabilizing its respective complex through protein-protein interactions.

Type I Hdr complexes have been shown to use $F_{420}H_2$ or formate as electron donors via FdhAB (17, 30), and our detailed analysis identifies that the type II HdrA complex could use $F_{420}H_2$ but not formate as an electron donor via FdhB and a non-catalytic Molybdop protein homologous to FdhA. This type II HdrA-Molybdop-FdhB gene cluster is a convergent evolutionary adaptation for the Hdr complex to use $F_{420}$ and is widely distributed among archaea involved in anaerobic methane and alkane metabolism (16) (Fig. 3).

Type III was another new type of HdrA identified in this study. We performed protein structure predictions and found the inserted GltD domain located near the HdrA's inserted Fd domain (Fig. 4). The Fd domain is used for Fd reduction in type I HdrA

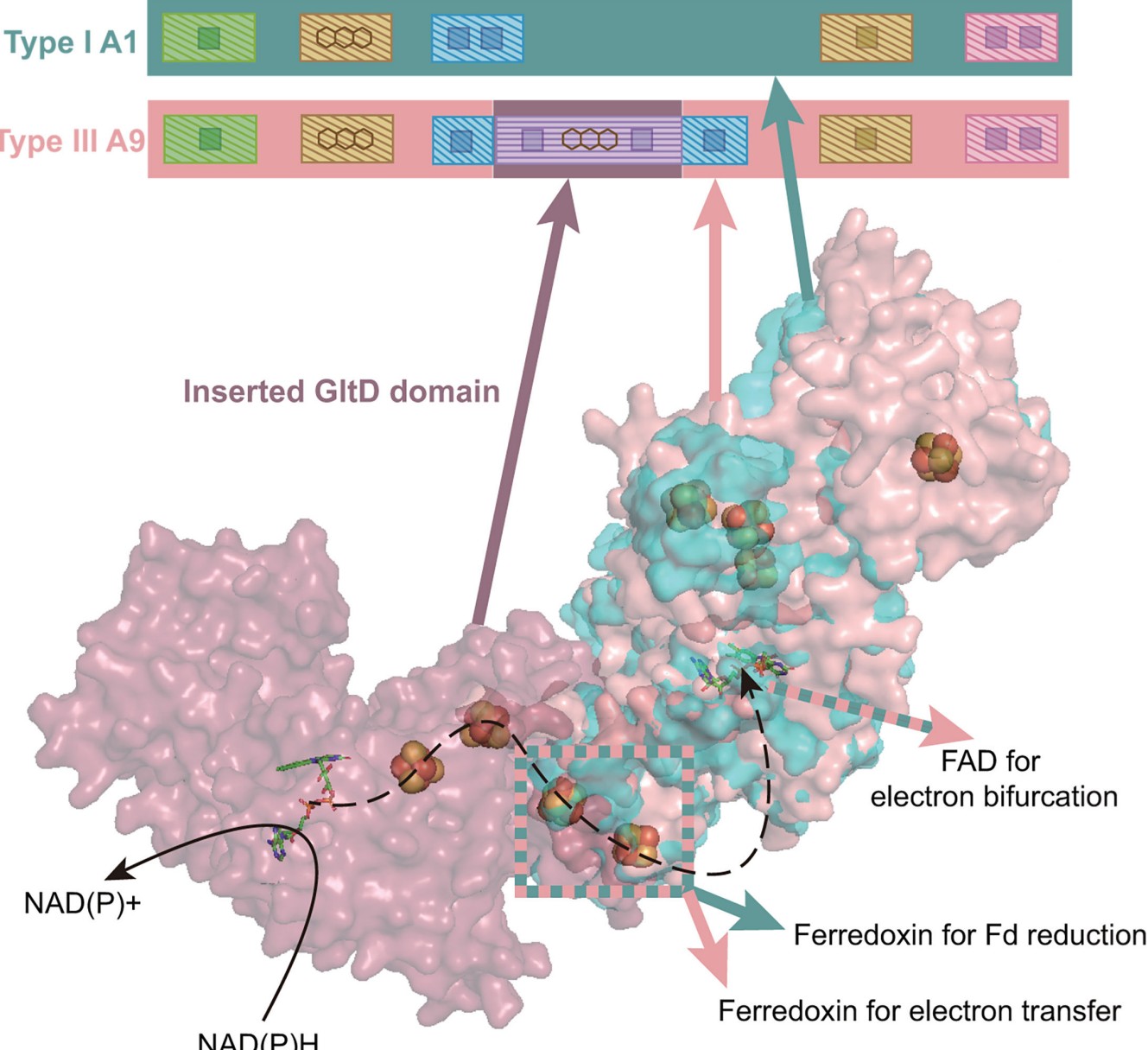

**FIG 4** Protein structure comparison of type III HdrA to type I HdrA. Part of type III A9 (pink) predicted structure can be aligned to characterized type I A1 (cyan) structure from *Methanothermobacter wolfeii*. The GltD domain in type III HdrA (dark pink), which is inserted between the Fd-reducing ferredoxin, may prevent type III HdrA from binding and reacting with Fd. Instead, the GltD domain could potentially react with NAD(P)H and transfer the electrons through a series of iron-sulfur clusters to the FAD for electron bifurcation. The schematic bar above the protein structures highlights the internal domains, with color-coded links between the schematic and the structural regions for clarity.

(24). However, in type III HdrA, the addition of the GltD domain could block the access of Fd to the inserted Fd domain (Fig. 4). Instead, the iron-sulfur clusters in Fd and GltD domains could be repurposed to transfer electrons from NADH instead of Fd. The gene cluster of type III *hdrA* in Bathyarchaeota contained *mvhADG*. We hypothesize that based on the reduction potentials of possible electron donors and acceptors, this type III HdrA complex might use NADH as its mid-potential electron donor for the reduction of CoB-S-S-CoM and $H^+$ to produce hydrogen, or the reverse FBEC reaction to produce NADH (Fig. 5). In ANME-1, type III *hdrA* was found in gene clusters with multiple *hdrA* but not with other functional genes, obscuring its cellular function. Type I HdrA complexes have been shown to use NADH as an electron donor indirectly through association with NADH dehydrogenases (48). However, the ability of the type III HdrA complex to directly utilize NADH could significantly improve electron transfer efficiency.

Lastly, by integrating data on neighboring genes along with the respective redox potentials of their substrates, we inferred the possible FBEB/FBEC reactions catalyzed by all four HdrA types (Fig. 5). Consistent with previous studies (13, 17, 24, 30), our gene cluster analysis showed that type I HdrA was versatile: it formed gene clusters with *mvhADG*, *fdhB*, and *fdhA* genes and therefore could use $H_2$, $F_{420}H_2$, or formate respectively as mid-potential electron donors or acceptors (Table S3). Also in gene clusters of type I *hdrA*, *hdrBC* genes were frequently identified as expected for the reduction of the high-potential electron acceptor CoB-S-S-CoM (24, 49), together with the ability of HdrA itself to reduce low-potential electron acceptor $Fd_{ox}$ or with the addition of *fmd/*

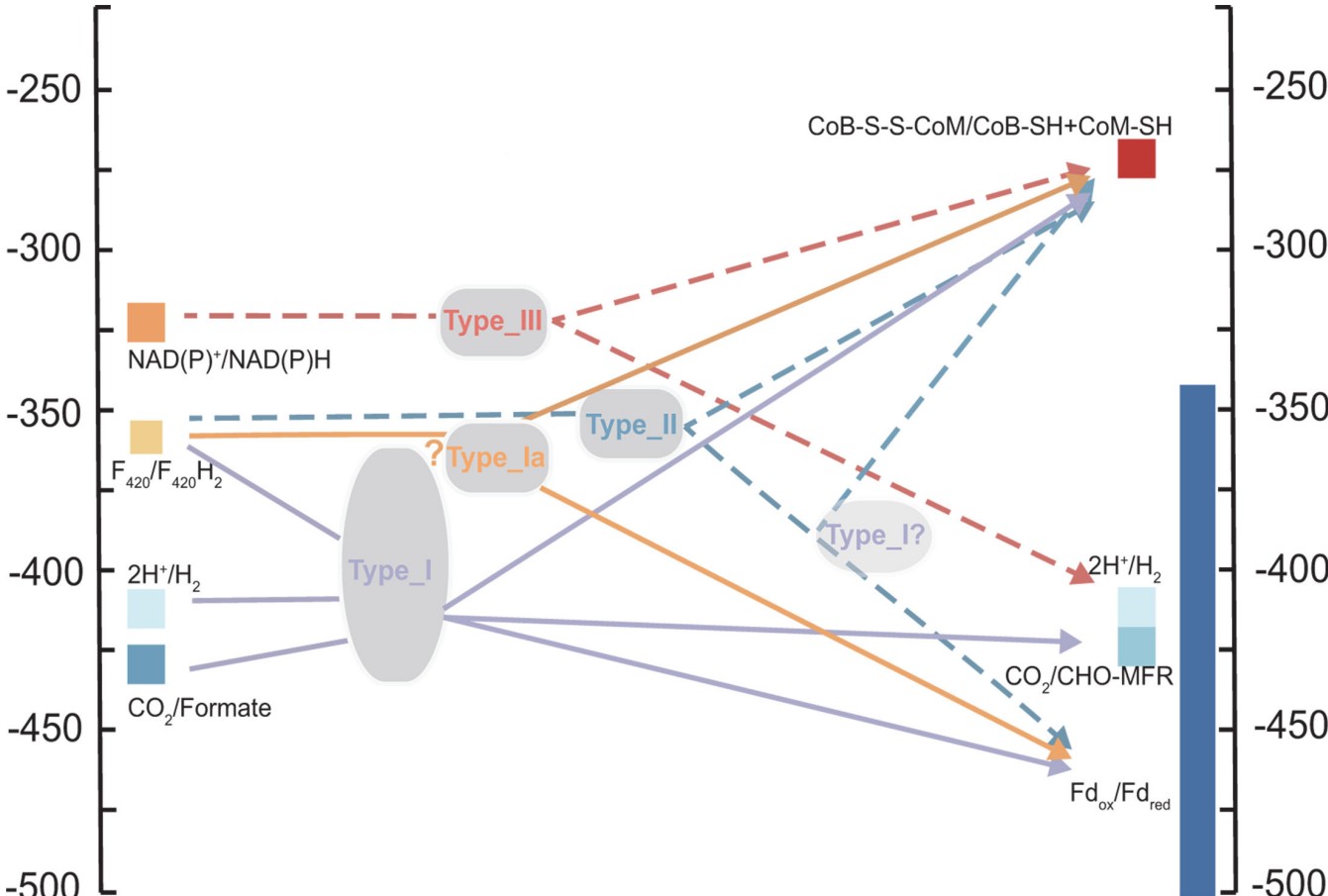

**FIG 5** Potential FBEB reactions of HdrA complexes in methane- and alkane-metabolizing archaea. Solid arrows represent experimentally validated FBEB reactions, while dashed arrows indicate reactions predicted in this study. Low-potential electrons from type II HdrA may be further bifurcated through a second type I HdrA that co-occurs in the same gene clusters. The "?" indicates the unresolved $F_{420}H_2$ oxidation mechanism by type Ia. While biochemical data indicate $F_{420}H_2$-dependent activity, type Ia lacks a conventional $F_{420}$-binding site, and the reaction mechanism is unclear.

*fwdABCDFG* genes to enable the Hdr complex to reduce $CO_2$ to formyl-methanofuran (Fig. 5; Table S3) (30). A previous study showed that type Ia HdrA protein alone, not in complex with $F_{420}$ dehydrogenase, could use $F_{420}H_2$ as its electron donor (17), despite lacking a recognizable $F_{420}$-binding site. Combining our analysis results of types II and III HdrA with previous characterization of type I and Ia HdrA (13, 20, 33, 50), we reveal how Hdr complexes have evolved distinct mechanistic solutions to catalyze even the same FBEB or FBEC reactions.

## DISCUSSION

HdrA proteins play a crucial role in energy conservation of methane- and alkane-metabolizing archaea. In the genomes of these archaea, the abundance of *hdrA* genes significantly exceeds that of *hdrBCDE* (Fig. S5), indicating a high functional and architectural diversity of HdrA complexes (40). The mid-point reduction potential of CoM-S-S-CoB was recently revised substantially lower from −138 mV to −281 mV (51), making FBEB reactions less energetically favorable and more remarkable in terms of enzyme catalysis. Through our analysis of 1,152 HdrA sequences together with their structural comparison, gene neighborhoods, and phylogeny, we provide a deeper understanding of how anaerobic microorganisms use HdrA to couple various electron carriers to the regeneration of their essential coenzymes B and M, while fulfilling their metabolic demands for low-potential electrons (Fig. 5).

As an ancient protein present in the earliest evolved methanogens, HdrA has been extensively modified and finetuned (9, 24, 52). Our categorization of HdrA reveals that 28 classes mostly vary in terms of the location and number of the iron-sulfur clusters (Fig. 1). These structural variations likely reflect adaptations to different interacting protein partners (24, 30, 48, 53). The lack of evolutionary pattern of these 28 classes suggests that the addition or deletion of the iron-sulfur cluster domains has been common and frequent.

Our categorization also distinguishes HdrA into four major types. Type I HdrA is considered the most traditional type and most contain both of the key residues, K409 and E356 (24, 39, 40) (Fig. 2). These two residues each serve an important function: K409 stabilizes negatively charged $FADH^-$ intermediate by interacting with N5 of FAD, and E356 maintains the position of K409 (24, 39). This is in contrast to TrxR family enzymes, where FAD binding primarily relies on non-specific backbone and water-mediated hydrogen bond interactions (39, 40). Type Ia HdrA evolved multiple times from type I by fusion with MvhD (Fig. 1). This fusion may improve electron transfer between MvhD and HdrA. In addition, the absence of partner HdrA in the heterotrimeric type Ia HdrABC (HdrA2B2C2) complex could create space near FAD to directly bind and react with $F_{420}$ (17, 40) (Fig. 5). Even after the removal of the fused MvhD, types I and Ia HdrA still form distinct evolutionary branches (Fig. 2; Fig. S2), suggesting that the fusion event occurred early in HdrA's evolution. No biochemistry investigation has been performed on the other two major types of HdrA. Type II HdrA contains two thioredoxin reductase domains, and both retain the two key amino acid residues for FBEB (Fig. 1 and 2), suggesting that type II HdrA has the capacity for two potential FBEB reactions within a single protein. Our phylogenetic analysis suggests that the N-terminus and C-terminus of type II HdrA were fused from two distinct type I HdrA classes (Fig. 2), rather than a gene duplication and subsequent fusion. This fusion likely occurred only once in the evolutionary history, as indicated by the clustering of all type II N-terminus separate from all type II C-terminus (Fig. 2). In contrast, type III HdrA only appears at the tip of the evolutionary branches nesting in two different phylogenetic clades, suggesting that the insertion of GltD into a type I HdrA occurred recently, or gene acquisition via horizontal gene transfer repurposed the Fd reduction site for the use of NADH. Our structural analysis of type III HdrA indicates a compelling example of functional adaptation through domain acquisition. The presence of the GltD domain with a GXGXXG motif is a hallmark of the Rossmann fold (35). The substitution of glycine in place of alanine at the final position of the motif optimally accommodates the adenine nucleoside of $NAD^+$ and

signifies a selection for $NAD^+$ binding over $NADP^+$ (36, 37). This adaptation suggests that type III HdrA has undergone functional specialization to adapt to cellular metabolic demands that favor $NAD^+$-dependent redox reactions (38).

In anaerobic methane-metabolizing archaea, a conserved gene cluster contains a previously uncharacterized Molybdop gene (homolog of FdhA) and $F_{420}$-dependent formate dehydrogenase beta subunit (FdhB or FrhB; Fig. 3). This gene cluster was first identified in ANME-2d genome (54) and hypothesized to perform electron confurcation from CoM-SH, CoB-SH, and $Fd_{red}$ to produce 2 $F_{420}H_2$ and later found to be conserved in methane- and alkane-oxidizing archaea (8, 16). However, the function of the Molybdop protein in these Hdr complexes remains unclear. Our detailed sequence and structural comparison reveal that this Molybdop protein is non-catalytic. The resulting HdrA complexes use $F_{420}H_2$ or $F_{420}$ via FdhB as their mid-potential electron donor or acceptor. The non-catalytic Molybdop protein may function as a structural scaffold within the Hdr complex, maintaining optimal positioning of iron-sulfur clusters and electron transfer routes in the protein complex. The presence and conservation of this gene cluster in all ANME-2 and ANME-3 suggest that this is an essential mechanism for $F_{420}H_2$ oxidation or production. Because the redox potential gap between $F_{420}H_2$ and Fd is larger than that between $H_2$ and Fd (55, 56), and because ANME possibly operates at a more oxidizing cellular redox state than methanogens (57, 58), additional energy may be needed to drive the reduction of low-potential Fd. This may be achieved by having a second HdrA in the complex together with an HdrD (Fig. 3A). The resulting soluble Hdr protein complex predicted from these gene clusters raises the possibility that a total of two FBEB reactions could lower the electron potential and drive the endergonic reduction of Fd (Fig. 5). This is not without precedence, as the class II benzoyl-CoA reductase complex has also been hypothesized to involve two FBEB reactions to lower the electron potential from NADH ($E' = -280$ mV) and drive the endergonic reduction of benzyl-CoA ($E^{0'} = -622$ mV) (40, 59).

While our results have shed light on the functions of HdrA in methane- and alkane-metabolizing archaea, questions still remain regarding the roles of HdrA proteins that are not in gene clusters but are present alone or together with other HdrA. Previous transcriptomic analyses suggest that these lone *hdrA* genes, including types I, Ia, and II, were expressed (Table S4) (60). Future biochemistry research and structural characterization could target these isolated HdrA variants, as well as types II and III HdrA that are in protein complexes with type I HdrA (Fig. 3 and 4). These experimental discoveries could provide important mechanistic insights into how anaerobes use FBEB/FBEC to conserve energy and thrive in diverse energy-limited environments.

## MATERIALS AND METHODS

### Collecting genomes of methane- and alkane-metabolizing archaea

Our genome analysis is based on 4,416 genomes from archaea species clusters in Genome Taxonomy Database (GTDB) v214 (61). Methane-metabolizing archaeal genomes containing *mcrABG* genes were identified using the Annotree (version RS 214) tool (62) based on its Kyoto Encyclopedia of Genes and Genomes (KEGG) and InterPro annotations (63–65). McrA was searched using KEGG entries K00399 and K00400 (McrA2) and InterPro entry IPR016212; McrB using KEGG entry K00401 and InterPro entry IPR003179; and McrG using KEGG entry K00402 and InterPro entry IPR003178. We manually added 14 alkane-metabolizing archaeal genomes discovered so far (6) in the GTDB v214 database. In total, 624 archaeal genomes were collected for subsequent analysis, representing a diverse range of species involved in methane and alkane metabolism.

## Identification and classification of HdrA

HdrA sequences from 624 genomes were identified using a combination of KEGG (entries K22480 and K03388) and InterPro (entry IPR0039650) annotation methods. Based on the previously reported shortest homologous HdrA sequence of 341 amino acids (39), sequences with fewer than 300 aa were filtered out as low quality or incomplete. The presence of a conserved hdrA domains (GOG1148, cl34141, or cl48997) in the remaining sequences was verified using NCBI Conserved Domain Search (CD-Search) (66, 67). As a result, a total of 1,152 HdrA sequences were identified in the genome collection.

Conserved domains in the HdrA sequences were analyzed using a combination of NCBI Conserved Domain Search (CD-Search) (66, 67) and manual identification of conserved motifs for iron-sulfur clusters and FAD-binding based on key residues identified in previous studies of HdrA protein structures (24) in their multiple sequence alignment (68). HdrA sequences were then classified based on the number and location of conserved domains in the protein sequence, including iron-sulfur cluster domain, thioredoxin reductase domain for FAD binding, and other functional domains such as MvhD and GltD.

## Sequence and structural analyses

The core region of the HdrA proteins was analyzed. To identify the core region, the collection of 1,153 HdrA sequences was first clustered using CD-HIT with a cutoff parameter of 80% (69), resulting in 401 representative sequences. For type II HdrA sequences, the C-terminal and N-terminal thioredoxin reductase domain were separated in their multiple sequence alignment into two sequences each containing a thioredoxin domain, labeled as type II_C and type II_N. For type III HdrA sequences, the inserted GltD domain region was identified based on comparison with type I HdrA sequences, and the inserted GltD domain was removed to avoid interfering with our phylogenetic analysis. The resulting type I, type Ia, type II_N, type II_C, and type III HdrA minus the inserted GltD domain sequences were aligned using ClustalOmega with parameters set to (--iter=5 --max-guidetree-iterations=5 --max-hmm-iterations=5 --threads=4) (68).

The core region was selected based on conserved amino acid residues essential for FAD binding and stabilization, ensuring conservation across all four HdrA types. Manual selection, referencing previous HdrA structures (24), identified 37 key residues within 4 Å of FAD, including those involved in hydrophobic interactions, hydrogen bonds, and water bridges. These amino acids collectively formed four complete sequence segments, totaling 133 amino acids. Additionally, conserved regions were extracted by automated selection from multiple sequence alignments. The aligned sequences were then trimmed using block mapping and gathering with entropy (BMGE) with the BLOSUM30 matrix (70). Both methods yielded highly similar conserved regions, and for enhanced reproducibility, the automated selection was ultimately chosen to define the core region consisting of 252 amino acids. The phylogenetic tree of the core regions of HdrA was then constructed using IQ-TREE (-m LG + G4 bb 1000 -bnni -nt AUTO) (71). The final tree was visualized using Interactive Tree of Life (iTOL) (72) with their metadata.

To investigate the gene neighborhood of the *hdrA* genes, Annoview v1.0 (73) was used to obtain the gene neighborhood containing the *hdrA* gene, and the gene clusters were manually identified based on gene transcription direction and gaps. The gene cluster information was combined with the HdrA categories to predict their potential functions.

To predict the structures of each type of HdrA and its associated protein complexes, AlphaFold3 (74) was utilized with the default parameters. The iron-sulfur clusters were integrated into the predicted structures using AlphaFill (75). The location of FAD was predicted by structural alignment with experimentally resolved structures available in the Protein Data Bank (PDB). The predicted protein structures were visualized using PyMOL version 3.0 (76). To investigate molybdopterin oxidoreductase proteins, a total of 85 sequences were extracted from our HdrA gene clusters and subjected to phylogenetic analysis along with sequences from the dimethyl sulfoxide reductase (MopB) family

metagenomic data sets (44). From the published MopB protein database, 709 sequences were selected as reference sequences using CD-HIT with a cutoff parameter of 50% to reduce redundancy (69). The phylogenetic analysis was performed in IQ-TREE as described above.

To further investigate the functions of unknown molybdopterin oxidoreductase proteins, the protein structures for FwdB, FhcB, FdhA, and FdhB were obtained from the PDB, with specific PDB IDs provided in the respective references (30, 46). Predicted structures using AlphaFold3 (74) and published structures were aligned in PyMOL version 3.0 (76). The key residues were identified in multiple sequence alignment using Clustal Omega and also visualized in PyMOL (68, 76).

The above bioinformatic workflow is also illustrated in Fig. S6 and summarized in supplementary file "Bioinformatics Tools and Workflows."

## ACKNOWLEDGMENTS

We sincerely thank Grayson Chadwick for his insightful discussions. This study was financially supported by the National Natural Science Foundation of China (92251305; 91951206; 42377109).

## AUTHOR AFFILIATION

[1]College of Urban and Environmental Sciences, Peking University, Beijing, China

## AUTHOR ORCIDs

Xingyu Lyu http://orcid.org/0009-0006-6050-2987
Hang Yu http://orcid.org/0000-0002-7600-1582
Yahai Lu http://orcid.org/0000-0002-1702-9868

## FUNDING

| Funder | Grant(s) | Author(s) |
| --- | --- | --- |
| National Natural Science Foundation of China | 92251305 | Xingyu Lyu |
| National Natural Science Foundation of China | 91951206 | Yahai Lu |
| National Natural Science Foundation of China | 42377109 | Hang Yu |

## ADDITIONAL FILES

The following material is available online.

### Supplemental Material

**Supplemental figures (Spectrum03238-24-s0001.docx).** Fig. S1 to S6.
**Supplemental material (Spectrum03238-24-s0002.txt).** Bioinformatics tools and workflows with detailed information.
**Supplemental tables (Spectrum03238-24-s0003.xlsx).** Tables S1 to S4.

### Open Peer Review

**PEER REVIEW HISTORY (review-history.pdf).** An accounting of the reviewer comments and feedback.

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
