## [Reviewer comments · Microbiology Spectrum]

Microbiology Spectrum

Diversity and Function of Soluble Heterodisulfide Reductases in Methane-Metabolizing Archaea

Xingyu Lyu, Hang Yu, and Yahai Lu

Corresponding Author(s): Xingyu Lyu, Peking University

Review Timeline:

Submission Date:	December 15, 2024
Editorial Decision:	January 15, 2025
Revision Received:	February 18, 2025
Accepted:	February 22, 2025

Editor: Zhe LYU

Reviewer(s): Disclosure of reviewer identity is with reference to reviewer comments included in decision letter(s). The following individuals involved in review of your submission have agreed to reveal their identity: Nana Shao (Reviewer #1); Kylie Allen (Reviewer #2)

Transaction Report:

DOI: <https://doi.org/10.1128/spectrum.03238-24>

Re: Spectrum03238-24 (Diversity and Function of Soluble Heterodisulfide Reductases in Methane-Metabolizing Archaea)

Dear Ms. Xingyu Lyu:

Thank you for the privilege of reviewing your work. Below you will find my comments, instructions from the Spectrum editorial office, and the reviewer comments.

Your manuscript has been reviewed by two experts. Both have showed enthusiasm towards it, although they have also raised some minor points. Please address these minor issues accordingly.

Revision Guidelines

Sincerely,
Zhe LYU
Editor
Microbiology Spectrum

Reviewer #1 (Comments for the Author):

In this manuscript, Lyu et al present a comprehensive and detailed analysis of 1,152 HdrA sequences from 624 genomes, incorporating structural comparison, gene neighborhood assessments, and phylogenetic analysis. They categorize HdrA into 28 distinct classes, which are further grouped into four major types: Type I, Ia, II, and III. Additionally, they also explore the

evolutionary relationships between these types and offer insights into their potential roles in electron bifurcation reactions. This is a high-quality manuscript, both in terms of figures and writing quality. This study provides valuable insights into the role of HdrA in methane- and alkane-metabolizing archaea, contributing to our understanding of its functional diversity. However, while the overall content is robust, the presentation of the findings could be improved. I have a few minor suggestions for enhancing the manuscript before publication.

Results:

- Figure 1 and Line 100, Please provide the names of the 28 HdrA classes and specify their source.
- Line 132-153, This description suggests that the authors consider moving Figure S1 to the main text in place of Figure 2. However, the colors in Figure S1 should be reorganized to enhance clarity and ensure a more concise presentation.
- Figure 4, The internal domains should be labeled in the figure, similar to the labeling in Figure 1.

Minor comment:

- Figure S2 is referenced in the text before Figure S1. Please revise the numbering of the supplementary figures accordingly.
- Figure 5, Please clarify the meaning of the "?" in the Type_Ia reactions.

Materials and methods:

- The code supporting the bioinformatics analyses should be made publicly available (e.g., on GitHub).
- Line 130-131, 349- 410, A workflow diagram, as presented in the supplementary data, would be very helpful for interpreting this work and guiding the reader through the analysis.

The discussion section didn't critically engage with existing literature. An improved presentation of this section would significantly improve the manuscript.

Reviewer #2 (Comments for the Author):

The manuscript by Xingyu Lyu et al. reports a comprehensive bioinformatic analysis of heterodisulfide reductase subunit A (HdrA) from methanogenic and methanotrophic archaea. They categorize HdrAs from 624 genomes into four major types based upon the presence and organization of distinct domains. They additionally perform structural predictions and analyses to probe the potential function of less-common Type II (in a gene cluster with MoCo oxidoreductase homolog) and Type III (containing a NAD(H)-binding module) HdrAs.

The information gleaned from these studies is highly valuable toward understanding the functional and evolutionary diversity of a key protein required for anaerobic methane metabolism and reveals very interesting targets for future biochemical and structural studies. Overall, the manuscript is well-written and well-presented. I only have minor points for the authors to consider.

1. Recommend to include a schematic of methanogenesis so that non-specialist readers can understand the function of Hdr within the context of the entire methanogenesis pathway.
2. There are many gene/protein names discussed throughout the manuscript and some have undefined or not clearly defined functions (i.e. MvhADG, FwdB, FdhA, FdhG, FhcB, FdhAB etc.). Please include a table with all of the genes/proteins discussed in the manuscript with the known functions summarized along with key references.
3. Lines 71-73 introduce FBEC (in ANME), but lacks references. Are there any reports of experimental evidence for FBEC, or is this simply implied from the reverse methanogenesis dogma? Please briefly discuss this in the introduction and include references as applicable.
4. At the end of the introduction, please explain the functions of HdrBC.
5. Line 92 - 65 genomes do not contain hdrA. Do these genomes only have hdrDE? Please briefly discuss this.
6. Lines 217-219 - how can the novel complex containing Molybdop and MvhD etc. be "highly similar" to previously characterized HdrABC-FdhAB complex since there are different proteins present? Please clarify.
7. Line 225-226 - the authors conclude that the Molybdop protein serves as non-catalytic but structurally supporting role on the basis of only computational analyses. Experimental work is needed to fully elucidate the function of the Molybdop protein. Thus, the authors must include wording like "we propose" . Additionally, how can the function be similar to FhcB if it is non-catalytic? Please explain this more completely.

Minor edits:

1. Line 21-22 of abstract - use "structural predictions" since the as-written wording makes it sound as though an experimental structure was obtained. Also, instead of "showed that" use "suggested that" since this idea will need further biochemical/structural studies to confirm.
2. Line 31 - change disulfide to heterodisulfide
3. Line 71 - please check that ref 15 is correct...it refers to bacterial Rnf complex while sentence is discussing hydrogenotrophic methanogens.
4. Line 161 - change "were lack of" to "lacked"
5. Line 181-184 - The presence of mvhD is indicated twice.
6. Line 238 - at the beginning of the new sentence, remove "In" and start with "The"
7. Line 278 - what is meant by "basal" methanogens? Consider using a different word here.
8. Line 329 - change "...that not in gene clusters.." to "...that are not in gene clusters..."
9. Line 332 - change "..isolate.." with "...isolated..."

10. Figure 4 legend description is too definitive when describing the function of GltD domain. Instead, it needs to use words like "may/could" or "potentially" when describing the possible function of GltD domain.

11. Figure S1 caption needs some edits "...for stabilizing FAD and residue." What is "and residue"? also, what is "notification" referring to in this context?

Responses to the Reviewers' comments

Diversity and Function of Soluble Heterodisulfide Reductases in

Methane-Metabolizing Archaea

Responses to Reviewer #1's comments

In this manuscript, Lyu et al present a comprehensive and detailed analysis of 1,152 HdrA sequences from 624 genomes, incorporating structural comparison, gene neighborhood assessments, and phylogenetic analysis. They categorize HdrA into 28 distinct classes, which are further grouped into four major types: Type I, Ia, II, and III. Additionally, they also explore the evolutionary relationships between these types and offer insights into their potential roles in electron bifurcation reactions. This is a high-quality manuscript, both in terms of figures and writing quality. This study provides valuable insights into the role of HdrA in methane- and alkane-metabolizing archaea, contributing to our understanding of its functional diversity. However, while the overall content is robust, the presentation of the findings could be improved. I have a few minor suggestions for enhancing the manuscript before publication.

Thank you for your positive comments and constructive feedback. We are encouraged by your recognition of the quality and scope of our analyses on HdrA's functional diversity. We carefully address each suggestion below to improve our manuscript.

Results:

1. Figure 1 and Line 100, Please provide the names of the 28 HdrA classes and specify their source.

We have addressed your suggestion by:

- L113 (corresponding to original PDF L100): We have added name of the HdrA classes to clarify the details.
- L116-123: We have explicitly specified the sources for these classes. The revised text now reads as follows:

“The HdrA classification system builds upon established nomenclature (16) and is structured around three primary types based on structural characteristics. Type I centers on A1, with A0 representing the minimal Hdr-like proteins in sulfur oxidizers, and variants named by domain modifications (e.g., A1-C lacks the C-terminal Fd domain). Type Ia derives from A2-1 (formerly HdrA2);(17,31,32), with subclasses defined by structural variations. Type II is founded on A3 and A4, while Type III is based on A9 and A10 (16), with their respective subclasses categorized by domain composition relative to these foundational structures.”

2. Line 132-153, This description suggests that the authors consider moving Figure S1 to the main text in place of Figure 2. However, the colors in Figure S1 should be reorganized to enhance clarity and ensure a more concise presentation.

We agree that the clarity of Figure S1 could be improved and made the following revisions:

- We have moved Figure S1 to the main text in place of Figure 2, as suggested.
- The colors in Figure S1 have been reorganized to enhance clarity and ensure a more concise presentation.

3. Figure 4, The internal domains should be labeled in the figure, similar to the labeling in Figure 1.

Figure 4 currently includes domain labels in the schematic bar above the protein structures, consistent with the labeling style used in Figure 1. To better highlight this feature, we have revised the figure legend to include the reference to these domain labels and their relationships:

Revised Figure Legend:

“**Figure 4.** ...The schematic bar above the protein structures highlights the internal domains, with color-coded links between the schematic and the structural regions for clarity.”

Minor comment:

1. Figure S2 is referenced in the text before Figure S1. Please revise the numbering of the supplementary figures accordingly.

Thank you for catching this discrepancy. We have reordered the supplementary figures so that their numbering matches their order of appearance in the text.

2. Figure 5, Please clarify the meaning of the "?" in the Type_Ia reactions.

Response:

The "?" indicates the mechanistic uncertainty of $F_{420}H_2$ oxidation by Type Ia. While previous work (Yan and Ferry, 2017) showed Type Ia HdrA can use $F_{420}H_2$ as electron donor, “?” denotes the lack of a conventional F_{420} -binding site.

We have revised the figure caption to clarify this point:

“**Figure 5.** ...The '?' indicates the unresolved $F_{420}H_2$ oxidation mechanism by Type Ia. While biochemical data indicates $F_{420}H_2$ -dependent activity, Type Ia lacks a conventional F_{420} -binding site and the reaction mechanism is unclear.”

Materials and methods:

1. The code supporting the bioinformatics analyses should be made publicly available (e.g., on GitHub).

Thank you for this suggestion. Our analysis primarily utilized standard bioinformatics tools and workflows. All specific parameters and software versions used in these analyses are fully documented in the Methods section, ensuring reproducibility of our work. Additionally, we have supplemented our work with detailed information in the supplementary file 'Bioinformatics Tools and Workflows' to facilitate the replication of our analyses.

2. Line 130-131,349- 410, A workflow diagram, as presented in the supplementary data, would be very helpful for interpreting this work and guiding the reader through the analysis.

To improve the clarity and interpretability of the analysis, we have created a workflow diagram summarizing the key steps of the bioinformatics analyses and their order. This diagram is now included as Figure S6 in the supplementary material.

3. The discussion section didn't critically engage with existing literature. An improved presentation of this section would significantly improve the manuscript.

We have thoroughly revised the Discussion section and included 8 new references to more critically engage with the existing literature and provide a deeper analysis of our findings in the context of previously published studies.

Responses to Reviewer #2's comments

The manuscript by Xingyu Lyu et al. reports a comprehensive bioinformatic analysis of heterodisulfide reductase subunit A (HdrA) from methanogenic and methanotrophic archaea. They categorize HdrAs from 624 genomes into four major types based upon the presence and organization of distinct domains. They additionally perform structural predictions and analyses to probe the potential function of less-common Type II (in a gene cluster with MoCo oxidoreductase homolog) and Type III (containing a NAD(H)-binding module) HdrAs.

The information gleaned from these studies is highly valuable toward understanding the functional and evolutionary diversity of a key protein required for anaerobic methane metabolism and reveals very interesting targets for future biochemical and structural studies. Overall, the manuscript is well-written and well-presented. I only have minor points for the authors to consider.

Thank you for your positive feedback and thorough evaluation of our work. We are encouraged by your recognition of our study's value in understanding HdrA's functional and evolutionary diversity in anaerobic methane metabolism. Your comment about the potential implications for future biochemical and structural studies aligns perfectly with our research goals. We will carefully address all your suggestions to further enhance the manuscript.

1. Recommend to include a schematic of methanogenesis so that non-specialist readers can understand the function of Hdr within the context of the entire methanogenesis pathway.

In the revised manuscript, we have included a schematic of the methanogenesis pathway to highlight the HdrA's critical role, now presented as Figure 1A.

2. There are many gene/protein names discussed throughout the manuscript and some have undefined or not clearly defined functions (i.e. MvhADG, FwdB, FdhA, FdhG, FhcB, FdhAB etc.). Please include a table with all of the genes/proteins discussed in the manuscript with the known functions summarized along with key references.

In response, we have created a comprehensive table summarizing all the genes and proteins discussed in the manuscript. This table has been added as Supplementary Table S3 providing readers with a clear and concise reference for the discussed genes/proteins and their functions.

3. Lines 71-73 introduce FBEC (in ANME), but lacks references. Are there any reports of experimental evidence for FBEC, or is this simply implied from the reverse methanogenesis dogma? Please briefly discuss this in the introduction and include references as applicable.

We have revised the manuscript to include relevant references and a brief discussion of FBEC in ANME. The existence of FBEC mechanisms in ANME is supported by comparative genomic studies and biochemical evidence. Chadwick et al. (2022) showed through comparative genomics that ANME possess the enzymes required for electron confurcation. Additionally, Yan and Ferry (2017) showed the reversibility of electron bifurcation to confurcation in a methanogen, providing a theoretical foundation for FBEC in reverse methanogenesis.

We have modified L70-72 (corresponding to original PDF L71-73):

“...as shown by comparative genomic and biochemical studies (16,17). This mechanism enables the coupling of endergonic and exergonic reactions in reverse methanogenesis.”

4. At the end of the introduction, please explain the functions of HdrBC.

We have added a description of the functions of HdrBC in the introduction section to explain of the role of HdrBC (L77–83): “In the complex, HdrB and HdrC form a functional unit responsible for catalysis and electron transfer. HdrB contains an unusual [4Fe-4S] cluster, serving as the key catalytic site for CoM-S-S-CoB reduction (24). HdrC facilitates electron transfer through its two conventional [4Fe-4S] clusters (25). The complete electron transfer pathway begins with HdrA receiving electrons from electron donors, transferring them via FAD and [4Fe-4S] clusters to HdrC, which then relays these electrons to HdrB's specialized [4Fe-4S] cluster for CoM-S-S-CoB reduction (3, 26).”

5. Line 92 - 65 genomes do not contain *hdrA*. Do these genomes only have *hdrDE*? Please briefly discuss this.

We have added a discussion on this in L98-106:

“Two methanogen groups, Methanonatronarchaeia and Verstraetearchaeota/Methanomethyliales, have been reported to lack HdrA and may utilize membrane-bound HdrD or HdrBC to reduce CoM-S-S-CoB (28, 29). Hydrogen-evolving membrane-bound hydrogenase (Mbh) complex or the energy-converting hydrogenase B (Ehb) complex could generate reduced ferredoxin required for anabolic reactions (8). In our analysis of other methanogens like Methanocaldococcus that lack HdrA, we identified partial *hdrA* sequences in their genomes. This absence of complete *hdrA* appears to result from the presence of selenocysteine residues, which are misinterpreted as stop codons in gene prediction tools.”

6. Lines 217-219 - how can the novel complex containing Molybdop and MvhD etc. be "highly similar" to previously characterized HdrABC-FdhAB complex since there are different proteins present? Please clarify.

We have clarified the basis for comparing the novel complex with the previously characterized HdrABC-FdhAB complex to explain the structural and functional similarities that underlie this comparison in the revised manuscript L237–241: “Although these complexes contain different protein components, their three-dimensional architectures share remarkable structural similarities, particularly in the core framework and arrangement of functional domains. PyMOL structural alignment shows that 8664 out of 9842 atoms could be superimposed with an RMSD of 3.911 Å (RMSD < 4 Å indicates significant structural similarity); (47).”

7. Line 225-226 - the authors conclude that the Molybdop protein serves as non-catalytic but structurally supporting role on the basis of only computational analyses. Experimental work is needed to fully elucidate the function of the Molybdop protein. Thus, the authors must include wording like "we propose". Additionally, how can the function be similar to FhcB if it is non-catalytic? Please explain this more completely.

We agree that our statement about needs to be more carefully worded. We have modified the text (Line 248-252) to explain the Molybdop protein's function and its functional similarity to FhcB: “Based on our structural analyses, we propose that the Molybdop protein serves as a non-catalytic but structurally supporting component in the protein complex. This proposed role is analogous to FhcB (46), which has been shown to lack catalytic activity but plays an essential structural role in stabilizing its respective complex through protein-protein interactions. ”.

Minor edits:

1. Line 21-22 of abstract - use "structural predictions" since the as-written wording makes it sound as though an experimental structure was obtained. Also, instead of "showed that" use "suggested that" since this idea will need further biochemical/structural studies to confirm.

We have modified the wording as suggested (L20).

2. Line 31 - change disulfide to heterodisulfide

We have modified the wording as suggested (L29).

3. Line 71 - please check that ref 15 is correct...it refers to bacterial Rnf complex while sentence is discussing hydrogenotrophic methanogens.

The updated reference has been replaced with the correct citation (L69).

4. Line 161 - change "were lack of" to "lacked"

We have changed “were lack of” to “lacked” as recommended (L181).

5. Line 181-184 - The presence of mvhD is indicated twice.

We have deleted the sentence regarding the presence of mvhD being indicated twice as suggested (L202-203).

6. Line 238 - at the beginning of the new sentence, remove "In" and start with "The"

We have modified the wording as suggested (L264).

7. Line 278 - what is meant by "basal" methanogens? Consider using a different word here.

We have revised the text (L308) by replacing “basal” with “earliest evolved” to better clarify the meaning.

8. Line 329 - change "...that not in gene clusters.." to "...that are not in gene clusters..."

We have modified the sentence as suggested (L378).

9. Line 332 - change "..isolate.." with "...isolated..."

We have modified the wording as suggested (L381).

10. Figure 4 legend description is too definitive when describing the function of GltD domain. Instead, it needs to use words like "may/could" or "potentially" when describing the possible function of GltD domain.

We have revised the Figure 4 legend to use "may" and "could potentially" to reflect the speculative nature of the GltD domain's function.

11. Figure S1 caption needs some edits "...for stabilizing FAD and residue." What is "and residue"? also, what is "notification" referring to in this context?

We have revised the Figure S1 caption to clarify the meaning. The updated caption now reads:

"The presence of key residues K409 and E356 for stabilizing FAD, along with other residues involved in FAD binding and stabilization. K409 and E356 refer to the corresponding residues in HdrA from *Methanothermococcus thermolithotrophicus* (WP_018154264)."

Re: Spectrum03238-24R1 (Diversity and Function of Soluble Heterodisulfide Reductases in Methane-Metabolizing Archaea)

Dear Ms. Xingyu Lyu:

Your manuscript has been accepted, and I am forwarding it to the ASM production staff for publication. Your paper will first be checked to make sure all elements meet the technical requirements. ASM staff will contact you if anything needs to be revised before copyediting and production can begin. Otherwise, you will be notified when your proofs are ready to be viewed.

Sincerely,
Zhe LYU
Editor
Microbiology Spectrum

Reviewer #1 (Comments for the Author):

The modifications introduced in the revised manuscript greatly improved the manuscript, and all of my comments have been answered.

Reviewer #2 (Comments for the Author):

The revised version is further improved based on the reviewers' feedback. I have no further suggestions.